# Nicotinic Acetylcholine Receptor Subunit Alpha-5 Promotes Radioresistance via Recruiting E2F Activity in Oral Squamous Cell Carcinoma

**DOI:** 10.3390/jcm8091454

**Published:** 2019-09-12

**Authors:** Che-Hsuan Lin, Hsun-Hua Lee, Chia-Hao Kuei, Hui-Yu Lin, Long-Sheng Lu, Fei-Peng Lee, Jungshan Chang, Jia-Yi Wang, Kai-Cheng Hsu, Yuan-Feng Lin

**Affiliations:** 1Graduate Institute of Medical Sciences, College of Medicine, Taipei Medical University (TMU), Taipei 11031, Taiwan; cloudfrank@gmail.com (C.-H.L.); js.chang@tmu.edu.tw (J.C.); 2Department of Otolaryngology, TMU Hospital, TMU, Taipei 11031, Taiwan; 3Graduate Institute of Clinical Medicine, College of Medicine, TMU, Taipei 11031, Taiwan; kaorulei@yahoo.com.tw (H.-H.L.); pplay1028@gmail.com (C.-H.K.); candycarol0227@gmail.com (H.-Y.L.); 4Department of Neurology, Shuang Ho Hospital, Taipei Medical University, New Taipei City 23561, Taiwan; 5Department of Neurology, School of Medicine, College of Medicine, TMU, Taipei 11031, Taiwan; 6Department of Neurology, Vertigo and Balance Impairment Center, Shuang Ho Hospital, TMU, New Taipei City 23561, Taiwan; 7Department of Urology, Division of Surgery, Cardinal Tien Hospital, Xindian District, New Taipei City 23148, Taiwan; 8Department of Breast Surgery and General Surgery, Division of Surgery, Cardinal Tien Hospital, Xindian District, New Taipei City 23148, Taiwan; 9Department of Radiation Oncology, TMU Hospital, TMU, Taipei 11031, Taiwan; lslu@tmu.edu.tw; 10Department of Otolaryngology, Shuang-Ho Hospital, TMU, New Taipei City 23561, Taiwan; fplee@tmu.edu.tw; 11Department of Otolaryngology, School of Medicine, College of Medicine, TMU, Taipei 11031, Taiwan; 12Graduate Institute of Cancer Biology and Drug Discovery, College of Medical Science and Technology, TMU, Taipei 11031, Taiwan; 13Cell Physiology and Molecular Image Research Center, Wan Fang Hospital, TMU, Taipei 11696, Taiwan

**Keywords:** radiotherapy, oral squamous cell carcinoma, *CHRNA5*, nicotinic acetylcholine receptor, E2F

## Abstract

Radiotherapy is commonly used to treat patients with oral squamous cell carcinoma (OSCC), but a subpopulation of OSCC patients shows a poor response to irradiation treatment. Therefore, identifying a biomarker to predict the effectiveness of radiotherapy in OSCC patients is urgently needed. In silico analysis of public databases revealed that upregulation of *CHRNA5*, the gene encoding nicotinic acetylcholine receptor subunit alpha-5, is extensively detected in primary tumors compared to normal tissues and predicts poor prognosis in OSCC patients. Moreover, *CHRNA5* transcript level was causally associated with the effective dose of irradiation in a panel of OSCC cell lines. Artificial silencing of *CHRNA5* expression enhanced, but nicotine reduced, the radiosensitivity of OSCC cells. Gene set enrichment analysis demonstrated that the E2F signaling pathway is highly activated in OSCC tissues with high levels of *CHRNA5* and in those derived from patients with cancer recurrence after radiotherapy. *CHRNA5* knockdown predominantly suppressed E2F activity and decreased the phosphorylation of the Rb protein; however, nicotine treatment dramatically promoted E2F activity and increased Rb phosphorylation, which was mitigated after *CHRNA5* knockdown in OSCC cells. Notably, the signature combining increased mRNA levels of *CHRNA5* and the E2F signaling gene set was associated with worse recurrence-free survival probability in OSCC patients recorded to be receiving radiotherapy. Our findings suggest that *CHRNA5* is not only a useful biomarker for predicting the effectiveness of radiotherapy but also a druggable target to enhance the cancericidal effect of irradiation on OSCC.

## 1. Introduction

Oral squamous cell carcinoma (OSCC) is the most predominant type of oral cancer and accounts for more than 90% of all oral cancers [1]. Based on an estimation by the Oral Cancer Foundation, 49,750 Americans were newly diagnosed with oral or oropharyngeal cancer in 2017, and 9750 individuals died from this cancer in 2017. The five-year survival rate of the newly diagnosed individuals is approximately 57%, and the disease-free survival rate is 58%. Unfortunately, the five-year survival rate has improved only slightly, from 50% to 57%, in decades. Oral habits such as betel quid chewing, alcohol consumption, and tobacco smoking have been documented as risk factors for oral cancer [2]. According to the National Comprehensive Cancer Network (NCCN) guidelines, treatment options vary depending on the cancer stage and include surgery, radiotherapy, chemotherapy, and targeted therapy (e.g., cetuximab) [3]. External beam radiotherapy (EBRT) is generally employed as the adjuvant to primary surgery to enhance loco-regional control in advanced cases or in cases with unfavorable pathological features, as a primary treatment for medically inoperable cases, including patients unable to tolerate or unsuited for surgery, and as a salvage or palliative treatment for persistent or recurrent disease [4]. Although modern radiotherapy plays an important role in the therapy of advanced head and neck cancers, including OSCC, further studies are needed to identify biomarkers that can precisely predict the effectiveness of radiotherapy in OSCC.

Nicotinic acetylcholine receptors (nAChRs) are ligand-gated ion channels that modulate key physiological processes ranging from neurotransmission to cancer signaling. Neuronal nAChRs are transmembrane proteins that form pentameric structures assembled from a family of subunits including α2–α10 and β2–β4 [5] and activated by the endogenous neurotransmitter acetylcholine (ACh), the exogenous tertiary alkaloid nicotine [6,7], and tobacco alkaloid [8]. Neuronal nAChR subunit alpha-5 (nAChRα5), a protein encoded by the *CHRNA5* gene, belongs to a superfamily of ligand-gated ion channels that mediate fast signal transmission at synapses. nAChRs are actively being investigated as drug targets for nervous system disorders, including Alzheimer’s disease, anxiety, attention-deficit hyperactivity disorder, depression, epilepsy, pain, Parkinson’s disease, schizophrenia, Tourette’s syndrome, and nicotine addiction [7,9,10]. To date, the *CHRNA5* rs16969968 polymorphism has been shown to be associated with the risk of delayed smoking cessation and increased risk of lung cancer among Caucasians [11,12], but not correlated with the risk of developing OSCC [13]. However, the prognostic significance of *CHRNA5* in OSCC progression (e.g., radioresistance) remains unknown. 

The aim of this study was thus focused on estimating the clinical relevance of *CHRNA5* and exploring the mechanism by which *CHRNA5* promotes radioresistance in OSCC. We found that higher *CHRNA5* transcript levels are extensively detected in primary tumors compared to normal tissues and significantly predict an increased risk for cancer recurrence after radiotherapy in OSCC patients. A cell-based radiosensitivity assay showed that upregulation or activation of *CHRNA5*-encoded neuronal nAChR increases the resistance of OSCC cells to irradiation (IR) treatment. Moreover, we found that the *CHRNA5*-promoted radioresistance in OSCC cells is probably related to accelerated cell cycle progression, as evidenced by the elevated level of phosphorylated Rb protein and enhanced activity of the E2F transcription factor. This study is the first to document that *CHRNA5* could be a prognostic factor in OSCC patients who elect to receive clinical radiotherapy. 

## 2. Materials and Methods

### 2.1. The Cancer Genome Atlas (TCGA)-Head and Neck Squamous Cell Carcinoma (HNSC) Patients and Data Processing

The clinicopathological data and transcriptional profiling of head and neck cancer patients were obtained from The Cancer Genome Atlas (TCGA) database and downloaded from UCSC Xena website (http://xena.ucsc.edu/welcome-to-ucsc-xena/) [14]. TCGA head and neck cancer patients who had complete clinicopathological information (Appendix A) were selected for this study. The RNA sequencing results of *CHRNA* members and E2F gene set deposited in The Molecular Signatures Database (http://software.broadinstitute.org/gsea/msigdb) [15] were downloaded from UCSC Xena website. The sum of mRNA levels obtained from the RNA sequencing results of E2F gene set in TCGA head and neck cancer patients was used to perform Pearson correlation test and Kaplan–Meier analyses. The stratification of mRNA levels regarding *CHRNA5* and E2F gene set into low- and high-level groups was determined by using Cutoff Finder (http://molpath.charite.de/cutoff/) [16] under the maximal risk condition in Kaplan–Meier analyses.

### 2.2. Cell Culture

The OSCC cell lines HSC-2, HSC-3, HSC-4, and SAS were cultured in Dulbecco’s modified Eagle’s medium (DMEM) containing 10% fetal bovine serum (FBS) and 1% nonessential amino acids (NEAA) at 37 ℃ in a humidified atmosphere containing 7% CO_2_. All cells were a gift of Dr. Michael Hsiao at Academia Sinica, Taiwan, and were authenticated on the basis of short tandem repeat (STR) analysis, morphologic and growth characteristics, and mycoplasma detection.

### 2.3. Reverse Transcription PCR (RT-PCR) and Quantitative PCR (Q-PCR)

Total RNA was extracted from cells using a TRIzol extraction kit (Invitrogen). Aliquots (5 μg) of total RNA were treated with Moloney-Murine Leukemia Virus (M-MLV) reverse transcriptase (Invitrogen) and amplified with Taqpolymerase (Protech) using paired primers (for *CHRNA5*, forward-CGCTCGATTCTATTCGCTACAT and reverse-CAGCACAGTCAAAGGATGAACTTT AC; for *GAPDH*, forward-AGGTCGGAGTCAACGGATTTG and reverse-GTGATGGCATG GACTGTGGTC) for PCR. Power SYBR^TM^ Green PCR Master Mix (Thermo Fisher) was used for Q-PCR. The mRNA levels were normalized to those of *GAPDH*. Fold changes were calculated using the 2^−ΔΔCt^ method.

### 2.4. Western Blotting Assay

Aliquots (20–100 μg) of total protein and HR Pre-Stained Protein Marker 10–170 kDa (BIOTOOLS, Taiwan) were loaded onto each lane of an SDS gel and transferred to PVDF membranes. The membranes were incubated with blocking buffer (5% nonfat milk in TBS containing 0.1% Tween-20) for 2 hours at room temperature. After that, the samples were incubated with primary antibodies against total/phosphorylated Rb (Cell Signaling) and GAPDH (AbFrontier) overnight at 4 °C. After extensive washing, the membranes were probed with secondary peroxidase-labeled IgGs for 1 hour at room temperature. Immunoreactive bands were visualized by enhanced chemiluminescence (Amersham Bioscience).

### 2.5. Preparation and Infection of Lentiviral Particle 

All derivatives of shRNA vectors with a puromycin selection marker (obtained from National RNAi Core Facility Platform in Taiwan) were transfected into the packaging cell line 293T (Thermo Fisher) along with the pMD.G and pCMVΔR8.91 plasmids, using a calcium phosphate transfection kit (Invitrogen). After 48 h incubation, the viral supernatants were collected and transferred to the target cells, and the infected cells were cultured in the presence of puromycin (Calbiochem) at 5–10 μg/mL in order to select stably transfected cells.

### 2.6. Irradiation Treatment and Cell Viability Analysis

Irradiation was performed with 6 MV X-rays using a linear accelerator (Digital M Mevatron Accelerator, Siemens Medical Systems, CA, USA) at a dose rate of 8 Gy/min. An additional 2 cm of a tissue-equivalent bolus material was placed on the tops of the plastic tissue culture flasks to ensure electronic equilibrium, and 10 cm of the tissue-equivalent material was placed under the flasks to obtain full backscatter. After irradiation treatment for 24 h, the cells were centrifuged and resuspended in an appropriate amount of PBS. For the cell viability assay, 20 μL of cell suspension was mixed with 20 μL of Trypan blue solution (0.4% in PBS). The stained cells were placed on a hemocytometer, and blue-stained cells were counted as nonviable under a microscope.

### 2.7. Luciferase Reporter Assay

Luciferase reporter vectors containing *E2F* response elements within the promoter region were purchased from Promega and utilized to estimate the activity of E2F. Cells were seeded in 6-well plates and cotransfected with firefly luciferase reporter and Renilla luciferase expression vectors. After 24 h, luciferase activity was measured using a Dual-Glo^®^ Luciferase Assay System (Promega). Briefly, the cells were lysed in lysis buffer containing a luciferase substrate for 10 min. Total lysis was achieved by centrifugation at 12,000 rpm for 1 min, and the supernatant was divided into three wells in a white 96-well plate to measure firefly luminescence. Dual-Glo^®^ Stop & Glo^®^ reagent was added to each well. After 10 min, Renilla luminescence was measured. The level of firefly luminescence was normalized to that of Renilla luminescence.

### 2.8. Statistical Analyses

SPSS 17.0 software (Informer Technologies, Roseau, Dominica) was used to analyze statistical significance. Student’s or paired samples t-tests were utilized to compare the gene expression of *CHRNA5* in a head and neck cancer cohort from TCGA. Spearman correlation was performed to estimate the association between *CHRNA5* mRNA expression and cell viability after irradiation exposure in the panel of OSCC cell lines. Pearson correlation analysis was used to evaluate the coexpression of *CHRNA5* and the E2F gene set in OSCC tissues. Survival probabilities were determined by Kaplan–Meier analysis and log-rank tests. A nonparametric Friedman test was used to analyze data from three or more related samples. In all analyses, *p* values <0.05 were considered statistically significant.

## 3. Results

### 3.1. CHRNA5 Upregulation Is Dominant in Primary Tumors Compared to Normal Tissues Derived from Patients with HNSCC

Since OSCC is included among HNSCCs, we first delineated the association of *CHRNA5* expression with tumorigenesis in HNSCC. Using TCGA HNSCC database, we performed transcriptional profiling of genes encoding nAChRs (Figure 1A). The mRNA levels of *CHRNA5*, *CHRNA6*, and *CHRNA9* in primary tumors were significantly (*p* < 0.001) higher than those in normal tissues (Figure 1B). In contrast, the transcript levels of *CHRNA2* and *CHRNA10* were shown to be dramatically decreased in primary tumors compared to normal tissues derived from HNSCC patients (Figure 1B). Moreover, the mRNA level of *CHRNA5* was increased more significantly (*p* = 3.2 x 10^-5^) than those of *CHRNA6* and *CHRNA9* in primary tumors compared to normal adjacent tissues derived from TCGA HNSCC patients (Figure 1C). Based on these findings, we sought to reveal the oncogenic role of *CHRNA5* in HNSCC.

### 3.2. CHRNA5 Upregulation Predicts a Poorer Prognosis in OSCC Patients Compared to Non-OSCC Patients

Because OSCC is the predominant type of HNSCC, we next estimated the clinical relevance of *CHRNA5* expression in OSCC and non-OSCC patients. By using the TCGA HNSCC database, we performed Kaplan–Meier analysis with respect to *CHRNA5* mRNA levels, HNSCC subdivision into OSCC or non-OSCC, and the signature obtained from the combination of *CHRNA5* mRNA levels with HNSCC subdivision to determine the overall survival (Figure 2A) or recurrence-free survival (RFS) (Figure 2B) probability. High *CHRNA5* transcript levels, OSCC type, and the combined signature of high *CHRNA5* mRNA levels with OSCC type were associated with poor prognosis regarding overall survival probability in HNSCC patients (Figure 2A). Regarding recurrence-free survival probability, high *CHRNA5* transcript levels, but not OSCC type, significantly predicted poor prognosis in patients with HNSCC (Figure 2B). Whereas the mean overall survival time was shortened in the OSCC cohort with higher *CHRNA5* mRNA levels compared to cohorts with other combined signatures (Figure 2A), the mean time to cancer recurrence did not differ between the OSCC and the non-OSCC cohorts with high *CHRNA5* transcript levels (Figure 2B). 

Furthermore, regarding recurrence-free survival, univariate and multivariate Cox regression analyses revealed that *CHRNA5* is an independent prognostic marker with respect to other clinical risk factors that predict cancer recurrence in HNSCC patients (Table 1). The chi-square test demonstrated that high *CHRNA5* transcript levels were significantly (*p* < 0.05) correlated with advanced pathological T stage and a record of being a smoker (Table 2). These results indicate that *CHRNA5* upregulation is associated with poor overall and recurrence-free survival probabilities in patients with HNSCC, particularly OSCC.

### 3.3. CHRNA5 Upregulation Is Associated with a Poor Response to Radiotherapy in OSCC Patients

Since *CHRNA5* upregulation was indicated to be an independent risk factor for cancer recurrence in HNSCC, we next performed another Kaplan–Meier analysis to examine the association with cancer recurrence in HNSCC patients who were recorded to be receiving radiotherapy. As shown in Figure 2C, high *CHRNA5* transcript levels were associated with a poor recurrence-free survival probability in HNSCC patients receiving radiation therapy. Moreover, the combined signature of high *CHRNA5* transcript levels with OSCC type predicted a worse prognosis in terms of recurrence-free survival probability in HNSCC patients receiving radiation therapy (Figure 2D). 

### 3.4. CHRNA5 Upregulation is Significantly Correlated with Radiosensitivity in OSCC Cells

We next examined the association between *CHRNA5* expression and radiosensitivity in a panel of OSCC cell lines. The mRNA levels measured by RT-PCR (Figure 3A) and quantitative PCR (Q-PCR) (Figure 3A) appeared to be relatively lower in HSC-3 cells than in other cell lines and were abundant in HSC-4 cells. Moreover, the cell viability after exposure to 8 Gy irradiation was dramatically reduced in HSC-3 cells but relatively higher in HSC-4 cells (Figure 3B). A significant (*p* < 0.0001) positive correlation was seen between *CHRNA5* expression and cell viability after exposure to 8 Gy irradiation in the examined OSCC cell lines (Figure 3C). Artificial silencing of *CHRNA5* (Figure 3D) dramatically enhanced the effectiveness of the irradiation treatment (Figure 3E) in HSC-4 cells. Conversely, treatment with nicotine dose-dependently promoted resistance of HSC-3 cells to the irradiation treatment (Figure 3F). 

### 3.5. CHRNA5 Upregulation/Activation Enhances E2F Activity in OSCC Cells

To determine the possible mechanism by which *CHRNA5* upregulation/activation confers radioresistance on OSCC cells, we analyzed the coexpression of *CHRNA5* and other somatic genes in TCGA OSCC subcohort with high *CHRNA5* transcript levels and cancer recurrence in the Kaplan–Meier analysis, by performing an in silico analysis using the gene set enrichment analysis (GSEA) software (Figure 4A). We then generated a gene set according to the results of Pearson correlation analysis with respect to the coexpression of *CHRNA5* and other somatic genes in the enrolled samples from TCGA OSCC dataset for GSEA simulation. The GSEA results strongly predicted E2F signaling pathway activation, as evidenced by the significantly (*p* < 0.001) positive enrichment score (ES) in the computational simulation against the generated gene set (Figure 4B,C). 

Since E2F activity is inhibited by nonphosphorylated tumor suppressor retinoblastoma (Rb) protein, we next examined the phosphorylation of the Rb protein in HSC-4 cells with or without *CHRNA5* knockdown. *CHRNA5* knockdown dramatically decreased the protein level of phosphorylated Rb (Figure 4D) and the activity of E2F (Figure 4E). Conversely, nicotine stimulation appeared to elevate the protein level of phosphorylated Rb (Figure 4F) and thereby enhance the activity of E2F (Figure 4G) in HSC-3 cells. 

### 3.6. Upregulation of the CHRNA5–E2F Axis Predicts Worse Prognosis in OSCC Patients Receiving Radiotherapy

Finally, we examined whether combining *CHRNA5* expression with enhanced E2F activity could more precisely predict irradiation effectiveness in OSCC patients. To this end, we obtained a gene set previously shown to reflect E2F activity from the Molecular Signatures Database (MSigDB, http://software.broadinstitute.org/gsea/msigdb/index.jsp) [15]. Pearson correlation analysis revealed that the mRNA levels of the E2F gene set and *CHRNA5* were significantly (*p* < 0.0001) positive in the examined TCGA OSCC tissues (Figure 5A). Kaplan–Meier analysis revealed that high expression levels of the E2F gene set was associated with poor prognosis in terms of recurrence-free survival probability in TCGA OSCC patients receiving radiotherapy (Figure 5B). In OSCC patients receiving radiotherapy, the gene signature of combined high levels of *CHRNA5* and E2F gene set appeared to predict worse prognosis in TCGA OSCC patients receiving radiotherapy (Figure 5C). 

## 4. Discussion

The identification of biomarkers to estimate the prognosis of patients electing to receive irradiation treatment is an important issue in the clinical control of OSCC. Here, we showed that *CHRNA5* might be a novel biomarker for predicting the therapeutic effectiveness of radiotherapy in OSCC patients. Nevertheless, further studies are needed to explore the clinical relevance of *CHRNA5* in the human papillomavirus (HPV)-related non-OSCC patients, since HPV has been thought to be a critical prognostic factor in non-OSCC tonsillar and base-of-tongue cancer [17]. Moreover, the expression of *CHRNA5* appeared to be positively correlated with the effective dose of irradiation in OSCC cell lines. In addition, nicotinic simulation via activation of *CHRNA5*-encoded neuronal nAChR increased the sensitivity of OSCC cells to irradiation treatment, probably owing to E2F-mediated acceleration of cell cycle progression. This study is the first to document that nAChRα5 and the E2F signaling axis confer radioresistance on OSCC cells. 

Nicotine stimulation has been shown to promote the acquisition of stem cell-like properties and epithelial–mesenchymal transition (EMT) features in HNSCC [18]. Moreover, exposure to nicotine has been found to enhance lymph node metastasis and cetuximab resistance in HNSCC [19]. In oral cancer cells, nicotine appeared to trigger EMT and promote cervical lymph node metastases in a mouse model [20]. In addition, activation of nAChR subunit alpha 7 by nicotine induced the proliferation of oral cancer cells [21]. Previous studies also demonstrated that nicotine exposure confers cisplatin resistance on nasal epithelial cancer [22] and oral cancer cells [23]. Previously, *CHRNA5* c.1192G >A polymorphism was found to be linked to frequency of tobacco chewing but did not have a significant impact on the risk of OSCC [13]. In this study, we further showed that nicotine exposure desensitizes oral cancer cells to irradiation treatment through the activation of *CHRNA5*-encoded neuronal nAChR. 

Induction of E2F signaling has been shown to be correlated with poor clinical outcome in oral cancer [24]. However, increeased Rb protein expression was detected in malignant lesions of OSCC [25]. Moreover, enforced expression of Rb in order to arrest cancer cells in the radiation-sensitive G2-M phase of the cell cycle has been found to enhance the effectiveness of irradiation in esophageal [26] and head and neck [27] squamous cell carcinomas. In lung cancer, nicotine exposure has been shown to induce E2F-regulated gene expression and thereby promote cell proliferation and cancer progression [28]. Here, we further showed that nicotine exposure enhances E2F activity by triggering Rb phosphorylation and ultimately confers radioresistance on OSCC cells, which is likely mediated by *CHRNA5*-encoded nAChR.

## 5. Conclusions

In summary, our results identify *CHRNA5* as a novel biomarker for predicting the therapeutic effectiveness of radiotherapy in OSCC patients. Activation of *CHRNA5*-encoded neuronal nAChR by its ligands, e.g., nicotine, may promote radioresistance by forcing cell cycle progression because of enhanced E2F activity. This finding, moreover, suggests the therapeutic targeting of *CHRNA5*-encoded nAChR as a new clinical strategy to enhance the cancericidal efficacy of irradiation in OSCC. 

## Figures and Tables

**Figure 1 jcm-08-01454-f001:**
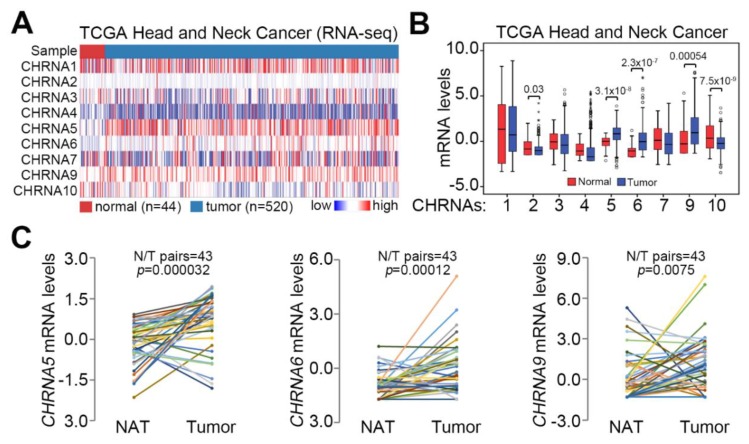
*CHRNA5* is upregulated in primary tumors compared to normal tissues derived from head and neck cancer patients. (**A**) Heatmap for the transcriptional profiling of genes encoding nicotinic acetylcholine receptors (nAChRs) using the head and neck cancer database in The Cancer Genome Atlas (TCGA). (**B**) Boxplot of the mRNA levels of nAChRs in normal tissues and primary tumors derived from the TCGA head and neck cancer database. Statistical differences were analyzed by the Student’s *t*-test. (**C**) The mRNA levels of *CHRNA5*, *CHRNA6*, and *CHRNA9* in normal adjacent tissues (NAT) and primary tumors from patients with head and neck cancer in the TCGA database. Statistical significance was evaluated by a paired samples *t-*test.

**Figure 2 jcm-08-01454-f002:**
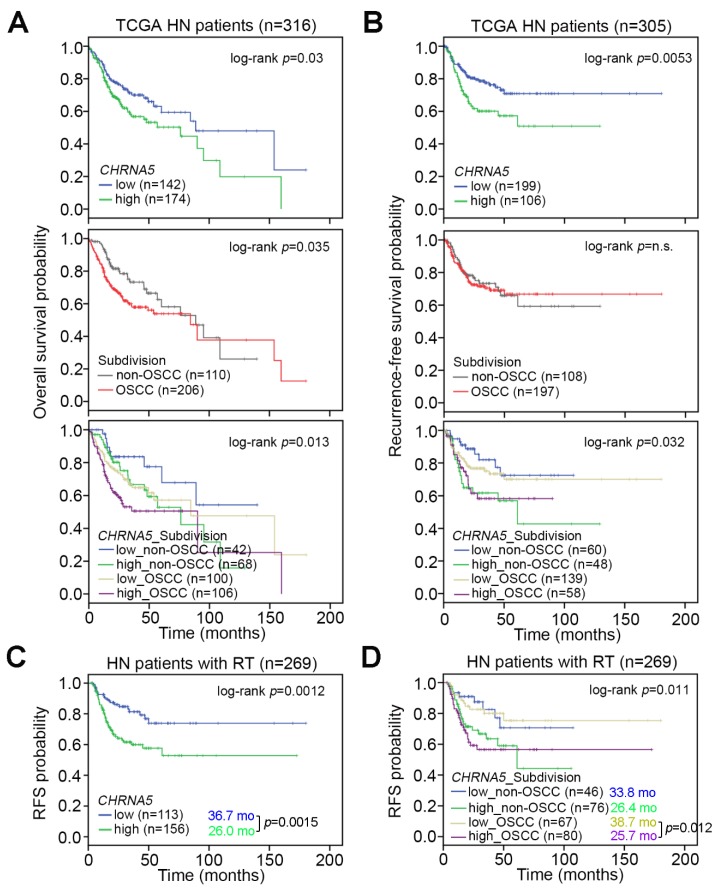
*CHRNA5* serves as a marker of poor prognosis in OSCC patients. (**A**) Kaplan–Meier analysis of overall survival probability for patients with head and neck cancer in TCGA database, with respect to *CHRNA5* expression (top), Head and Neck Squamous Cell Carcinoma (HNSCC) subdivision (middle,) and *CHRNA5* expression combined with HNSCC subdivision (bottom). (**B**) Kaplan–Meier analysis of recurrence-free survival probability for patients with head and neck cancer in TCGA database, with respect to *CHRNA5* expression (top), HNSCC subdivision (middle), and *CHRNA5* expression combined with HNSCC subdivision (bottom). (C and D) Kaplan–Meier analysis of recurrence-free survival (RFS) probability for HNSCC patients treated with radiotherapy, with respect to *CHRNA5* mRNA levels (**C**) and the combined signature of *CHRNA5* mRNA levels with HNSCC subdivision (**D**). In (**C**,**D**), the statistical difference in the average survival time was analyzed by Student’s *t-*test.

**Figure 3 jcm-08-01454-f003:**
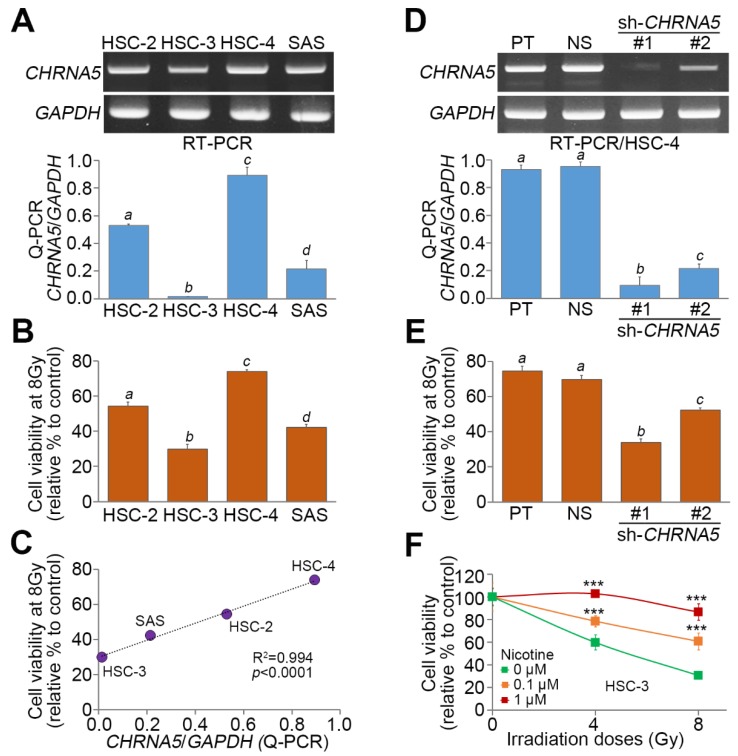
Endogenous *CHRNA5* expression and radiosensitivity in OSCC cells. (**A**) mRNA levels of *CHRNA5*, as measured by RT-PCR (upper) and Q-PCR (lower), in a panel of OSCC cell lines (HSC-2, HSC-3, HSC-4, and SAS). (**B**) Viability of the examined OSCC cells at 24 h after exposure to 8 Gy irradiation. (**C**) Doplot for the correlation between *CHRNA5* expression and cell viability at 24 h after exposure to 8 Gy irradiation in the examined OSCC cell lines. (**D**) mRNA levels of *CHRNA5,* as measured by RT-PCR (upper) and Q-PCR (lower), in HSC-4 cells transfected without (parental, PT) or with nonsilencing (NS) control shRNA or with two independent shRNA clones targeting *CHRNA5*. (**E**) Viability of the detected HSC-4 cell variants at 24 h after exposure to 8 Gy irradiation. (**F**) Viability of HSC-3 cells pretreated with nicotine at the indicated concentrations for 24 h, measured at 24 h after exposure to the designated doses of irradiation. In (**A**,**B**,**D**–**F**), the error bars denote the data from three independent experiments presented as the mean ± SEM, and the different letters indicate the statistical significance at *p* < 0.001 estimated by the nonparametric Friedman test. In (**F**), the symbol “***” denotes *p* < 0.0001.

**Figure 4 jcm-08-01454-f004:**
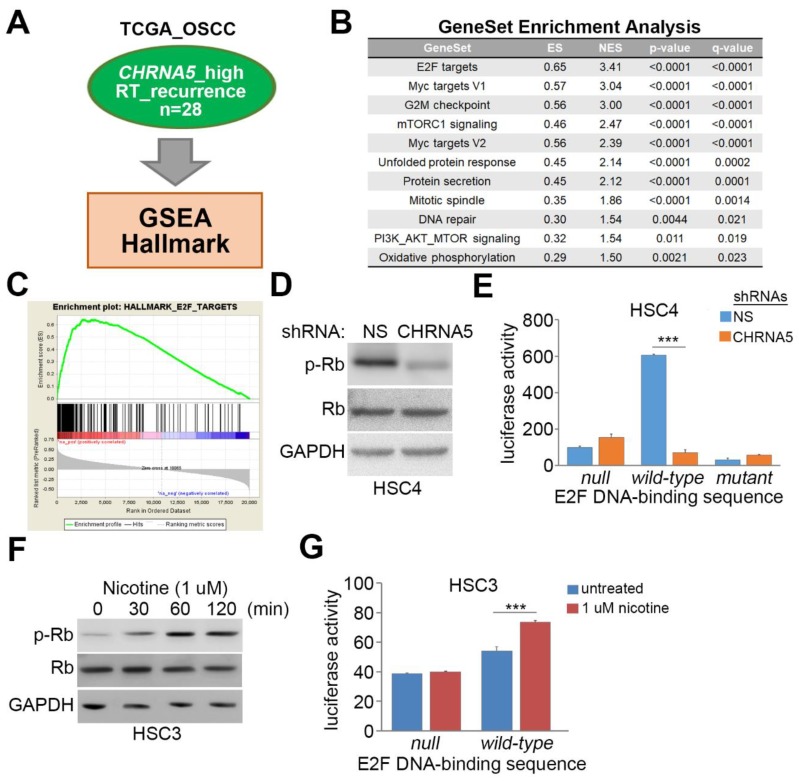
CHRNA5 activation triggers E2F activity in OSCC cells. (**A**) Flowchart of gene set enrichment analysis (GSEA) with respect to the Pearson correlation coefficient (r) derived from the correlation among *CHRNA5* transcripts and coexpressed somatic genes. (**B**) Gene sets predicted to be significantly correlated with the *CHRNA5* coexpression signature derived from OSCC tissues in TCGA as indicated in A. The parameters including enrichment score (ES), normalized enrichment score (NES), nominal *p* value, and false discovery rate (FDR) are shown. (**C**) The ES derived from the correlation between the E2F signaling gene signature and the queried Pearson correlation coefficient is plotted (green curve). (**D**) Western blot analysis for phosphorylated retinoblastoma (p-Rb), Rb, and GAPDH proteins derived from HSC-4 cells transfected with nonsilencing (NS) or *CHRNA5* shRNA. (**E**) Luciferase activity detected in NS or *CHRNA5*-silenced HSC-4 cells transfected with a luciferase expression vector without (null) or with wild-type or mutant *E2F* DNA-binding sequences. (**F**) Western blot analysis for p-Rb, Rb, and GAPDH proteins derived from HSC-3 cells treated with 1 µM nicotine for the indicated time periods. (**G**) Luciferase activity detected in HSC-3 cells transfected with a luciferase expression vector without (null) or with the wild-type *E2F* DNA-binding sequence and treated without (untreated) or with 1 µM nicotine for 24 h. In (**D**,**F**), GAPDH was used as the internal control for protein loading. In (**E**,**G**), the error bars denote the data from three independent experiments presented as the mean ± SEM, and the symbol “***” indicates the statistical significance at *p* < 0.001, as estimated by the nonparametric Friedman test.

**Figure 5 jcm-08-01454-f005:**
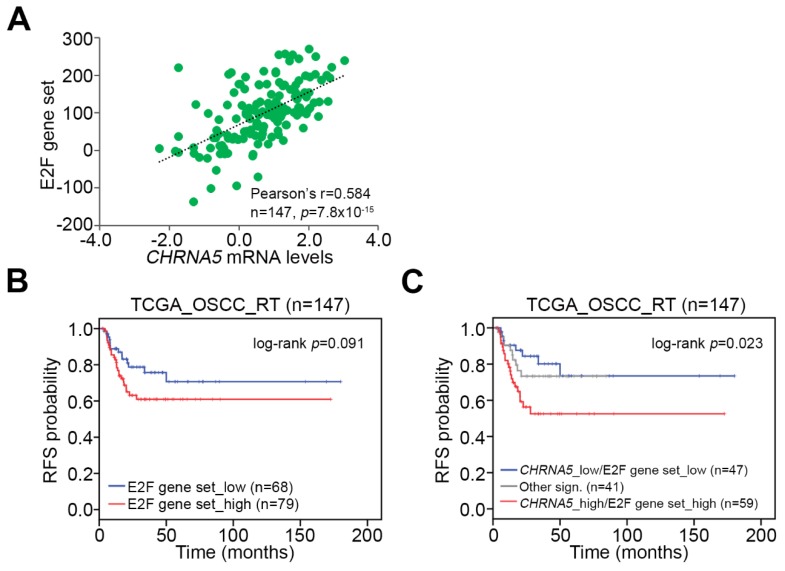
Prognostic significance of *CHRNA5* transcript levels combined with the E2F gene signature in OSCC patients receiving radiotherapy. (**A**) Pearson correlation analysis for the sum of mRNA levels of E2F gene set and *CHRNA5* mRNA levels in TCGA OSCC tissues. (B and C) Kaplan–Meier analyses for TCGA OSCC patients receiving radiotherapy, stratified by the transcript levels of the E2F gene set alone (**B**) or combined with the mRNA levels of *CHRNA5* (C). In C, other signatures (sign.) indicate the combination of low-level *CHRNA5* (*CHRNA5*_low) with low-level E2F gene set and of high-level *CHRNA5* with high-level E2F gene set.

**Table 1 jcm-08-01454-t001:** Cox univariate and multivariate analyses of recurrence-free survival probability in association with *CHRNA5* mRNA expression levels and pathological stage in a TCGA cohort with head and neck cancers. The abbreviations OSCC, pT, pN, RT and TarT stand for oral squamous cell carcinoma, pathologic T stage, pathologic N stage, radiation theray and targeted therapy, respectively.

Recurrence-free survival (*n* = 305)				
Variable	Crude HR (95% CI)	*p*	Adjusted HR (95% CI)	*p*
**age**				
< 60	1	NA	1	NA
≥ 60	1.27 (0.81–1.98)	0.301	1.46 (0.91–2.36)	0.117
**gender**				
female	1	NA	1	NA
male	1.08 (0.64–1.81)	0.781	1.06 (0.61–1.84)	0.842
**pT**				
T1–T2	1	NA	1	NA
T3–T4	1.82 (1.11–2.98)	0.018	1.47 (0.87–2.49)	0.150
**pN**				
N0	1	NA	1	NA
N1–N3	1.72 (1.07–2.76)	0.024	1.55 (0.92–2.259)	0.097
**stage**				
I–II	1	NA	NA	NA
III–IV	2.38 (1.19–4.76)	0.015	NA	NA
**treatment**				
none	1	NA	1	NA
RT	1.27 (0.75–2.28)	0.428	1.04 (0.56–1.92)	0.909
RT + TarT	1.52 (0.86–2.68)	0.150	1.22 (0.63–2.36)	0.549
**smoker**				
no	1	NA	1	NA
yes	1.32 (0.75–2.32)	0.337	1.10 (0.60–2.03)	0.762
**subdivision**				
non-OSCC	1	NA	1	NA
OSCC	1.04 (0.66–1.66)	0.855	1.28 (0.78–2.10)	0.321
***CHRNA5* level**				
low	1	NA	1	NA
high	1.87 (1.20–2.91)	0.006	1.82 (1.15–2.88)	0.011

**Table 2 jcm-08-01454-t002:** Relationship between *CHRNA5* expression and the clinicopathological characteristics of the TCGA cohort with head and neck cancers.

Clinicopathological Characteristics	*n*	*CHRNA5* Expression, *n* (%)		*p*
		Low (*n* = 199)	High (*n* = 106)	
**age**				
< 60	151	92 (60.9%)	59 (39.1%)	
≥ 60	154	107 (69.5%)	47 (30.5%)	0.120
**gender**				
female	82	57 (69.5%)	25 (30.5%)	
male	223	142 (63.7%)	81 (36.3%)	0.416
**pT**				
T1–T2	119	89 (74.8%)	30 (25.2%)	
T3–T4	186	110 (59.1%)	76 (40.9%)	0.007
**pN**				
N0	133	95 (71.4%)	38 (28.6%)	
N1–N3	172	104 (60.5%)	68 (39.5%)	0.053
**stage**				
I–II	63	49 (77.8%)	14 (22.2%)	
III–IV	242	150 (62.0%)	92 (38.0%)	0.019
**smoker**				
no	72	55 (76.4%)	17 (23.6%)	
yes	233	144 (61.8%)	89 (38.2%)	0.024
**subdivision**				
non-OSCC	108	60 (55.6%)	48 (44.4%)	
OSCC	197	139 (70.6%)	58 (29.4%)	0.012

*p* values were derived with a two-sided Pearson chi-square test.

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
