# Peer review of "Nicotinic Acetylcholine Receptor Subunit Alpha-5 Promotes Radioresistance via Recruiting E2F Activity in Oral Squamous Cell Carcinoma"

_jcm, 2019, doi:10.3390/jcm8091454_

Round 1

Reviewer 1 Report

Nicotinic acetylcholine receptor subunit alpha-5 promotes radioresistance via recruiting E2F activity in oral squamous cell carcinoma, by Lin CH et al.

In this study authors do not precisely state its purpose. Instead of that, they describe some findings of this study. The aim should be clearly defined at the end of the Introduction.

Although the findings seem to be interesting, enthusiasm is dampened by some concerns. 

In the Materials and Methods (M&M) section, are described methods such as cell culture, RT-PCR, irradiation of cell cultures, luciferase assay and statistical methods. No patients are presented here. However, in the Results section survival data are presented. The number of cases, their selection and their clinicopathological characteristics should be precisely described in M&M section. And exactly the same should be done for the controls. Who are exactly the non-OSCC patients? What are their characteristics? It would be helpful to know if all cancer patients were treated only with surgery and details of that surgery. Where any of the patients treated with radiation therapy or adjuvant chemotherapy?  On Table 1 an univariate and multivariate survival analysis was performed in a cohort with head and neck cancers. According to the title the study address the oral squamous cell carcinoma. Thus, in this table there are cancers of other locations. Or not? I wonder how many of these were HPV related? Clearly if they were base of tongue this could be true and a confounder. This is not discussed.  On Table 1, in the adjusted HR column, N and clinical stage are not significantly related with survival. This finding contradicts with a number of other clinical studies on oral squamous cell carcinomas, which show N and clinical stage are associated with worse prognosis or more aggressive phenotypes. The authors should acknowledge these published opposing studies.  Line 182, 3.3. CHRNA5 upregulation probably correlates … What do authors want to say using the term “probably”. Are they sure of the meaning of their results? Table 2 also refers to head and neck cancers instead of OSCC. Why do not authors limit its study to oral squamous cell carcinoma (OSCC) as the title proclaims. There are some typos in the text. For example: line 174 “assoaicates”; Figure 2 legend, lines 195 and 197, head and nech; line 209 “significaly”; line 284, “treatempt”; line 285 “thereapeutic”.

Author Response

Reviewer 1:

In this study authors do not precisely state its purpose. Instead of that, they describe some findings of this study. The aim should be clearly defined at the end of the Introduction.

Although the findings seem to be interesting, enthusiasm is dampened by some concerns. 

In the Materials and Methods (M&M) section, are described methods such as cell culture, RT-PCR, irradiation of cell cultures, luciferase assay and statistical methods. No patients are presented here. However, in the Results section survival data are presented. The number of cases, their selection and their clinicopathological characteristics should be precisely described in M&M section. And exactly the same should be done for the controls. Who are exactly the non-OSCC patients? What are their characteristics? It would be helpful to know if all cancer patients were treated only with surgery and details of that surgery. Where any of the patients treated with radiation therapy or adjuvant chemotherapy? On Table 1 an univariate and multivariate survival analysis was performed in a cohort with head and neck cancers. According to the title the study address the oral squamous cell carcinoma. Thus, in this table there are cancers of other locations. Or not? I wonder how many of these were HPV related? Clearly if they were base of tongue this could be true and a confounder. This is not discussed.  On Table 1, in the adjusted HR column, N and clinical stage are not significantly related with survival. This finding contradicts with a number of other clinical studies on oral squamous cell carcinomas, which show N and clinical stage are associated with worse prognosis or more aggressive phenotypes. The authors should acknowledge these published opposing studies.  Line 182, 3.3. CHRNA5 upregulation probably correlates … What do authors want to say using the term “probably”. Are they sure of the meaning of their results? Table 2 also refers to head and neck cancers instead of OSCC. Why do not authors limit its study to oral squamous cell carcinoma (OSCC) as the title proclaims. There are some typos in the text. For example: line 174 “assoaicates”; Figure 2 legend, lines 195 and 197, head and nech; line 209 “significaly”; line 284, “treatempt”; line 285 “thereapeutic”.

Point-by-point responses

Q1: In this study authors do not precisely state its purpose. Instead of that, they describe some findings of this study. The aim should be clearly defined at the end of the Introduction.

Re: We sincerely thank the Reviewer for these constructive comments and suggestions. As suggested, we have added sentences “The aim of this study was thus focused on estimating the clinical relevance of CHRNA5 and exploring the mechanism in which CHRNA5 promotes radioresistance in OSCC” to clearly define our purpose at the end of the Introduction (Please see Lines 85-86 of the revision).

Q2: In the Materials and Methods (M&M) section, are described methods such as cell culture, RT-PCR, irradiation of cell cultures, luciferase assay and statistical methods. No patients are presented here. However, in the Results section survival data are presented. The number of cases, their selection and their clinicopathological characteristics should be precisely described in M&M section. And exactly the same should be done for the controls. Who are exactly the non-OSCC patients? What are their characteristics? It would be helpful to know if all cancer patients were treated only with surgery and details of that surgery. Where any of the patients treated with radiation therapy or adjuvant chemotherapy?

Re: As requested, we have added a paragraph in M&M section and the Supplementary Table 1 of revision to describe the information of TCGA head and neck cancer patients. The content of the paragraph is described as follow.

2.1 TCGA-HNSC patients and data processing

   The clinicopathological data and transcriptional profiling of head and neck cancer patients were obtained from The Cancer Genome Atlas (TCGA) database and downloaded from UCSC Xena website (http://xena.ucsc.edu/welcome-to-ucsc-xena/). The TCGA head and neck cancer patients who have complete clinicopathological information (Tab. S1) were selected in this study. The RNA sequencing results of CHRNA members and E2F geneset deposited in The Molecular Signatures Database (http://software.broadinstitute.org/gsea/msigdb) were downloaded from UCSC Xena website. The E2F signature was generated by using the sum of mRNA levels obtained from the RNA sequencing results of E2F geneset in TCGA head and neck cancer patients. The definition of low and high-levels of CHRNA5 and E2F signature was determined by using Cutoff Finder (http://molpath.charite.de/cutoff/) under the maximal risk condition in Kaplan-Meier analyses.

Q3: On Table 1 an univariate and multivariate survival analysis was performed in a cohort with head and neck cancers. According to the title the study address the oral squamous cell carcinoma. Thus, in this table there are cancers of other locations. Or not? I wonder how many of these were HPV related? Clearly if they were base of tongue this could be true and a confounder. This is not discussed.  On Table 1, in the adjusted HR column, N and clinical stage are not significantly related with survival. This finding contradicts with a number of other clinical studies on oral squamous cell carcinomas, which show N and clinical stage are associated with worse prognosis or more aggressive phenotypes. The authors should acknowledge these published opposing studies. 

Re: Thanks for these critical comments. On table 1, the locations of TCGA-HNSC are listed in the Supplementary Table 1. There are 11 TCGA-HNSC tissues originated from base of tongue in the tested cohort on table 1 and classified as non-OSCC type in this study. According to TCGA information, the 4 (TCGA-BB-7861-01, TCGA-CN-A6UY-01, TCGA-CN-A6V6-01 and TCGA-HD-8314-01) of 11 TCGA-HNSC tissues originated from base of tongue are positive for HPV determined by p16 testing. Indeed, HPV was recognized by the International Agency for Research on Cancer as a risk factor for oropharyngeal squamous cell carcinoma, where tonsillar and base of tongue cancer (TSCC and BOTSCC) dominate. Furthermore, patients with HPV-positive TSCC and BOTSCC, had a much better clinical outcome than those with corresponding HPV-negative cancer and other head and neck cancer. On Table 1, as compared to non-OSCC including TSCC (n=15) and BOTSCC (n=11), OSCC type serves as a risk factor for cancer recurrence in TCGA-HNSC patients even though the results are not significant in Cox regression test. Nevertheless, in Figure 2A, the overall survival time in TCGA-HNSC patients with OSCC is significantly (p=0.035) shorter than that of non-OSCC. Moreover, TCGA-HNSC patients with OSCC harboring a high-level CHRNA5 show a worse overall survival. Since we were interested in the prognostic significance of CHRNA5 in OSCC patients, the results of HPV detection was not discussed in the old version.

   As recommended, we added the sentences “Nevertheless, further studies are still needed to explore the clinical relevance of CHRNA5 in the human papillomavirus (HPV)-related non-OSCC patients since HPV has been thought to be a critical prognostic factor in the non-OSCC tonsillar and base of tongue cancer [14].” to discuss this issue in the Discussion section of revision (Please see Lines 319-322 of revision). We also provided this report as Ref.14 in the new Reference section as follow.

T. Ramqvist, N. Grün, and T. Dalianis. Human papillomavirus and tonsillar and base of tongue cancer. Viruses. 7 (2015) 1332–1343.

   On Table 1, the adjusted HR was estimated by Cox regression test using a multivariable analysis mode against the listed risk factors under recurrence-free survival condition. In comparison with other confounders, indeed, N and clinical stage are not the independent prognostic factor even though they are risk factors in a univariable analysis under recurrence-free survival condition. However, in Kaplan-Meier survival analysis, lymph node-positive (N1-N3) and higher clinical stage (III-IV) are associated with poor recurrence-free survival probability in TCGA-OSCC patients (Please see following figures). Therefore, our findings are similar to those previous reports that show N and clinical stage are associated with worse prognosis or more aggressive phenotypes in OSCC.

Q4: Line 182, 3.3. CHRNA5 upregulation probably correlates … What do authors want to say using the term “probably”. Are they sure of the meaning of their results?

Re: Thank you for this critical comment. We have changed this subtitle “3.3. CHRNA5 upregulation probably correlates with the mechanism for radiation and targeted drug resistance in OSCC” to “3.3. CHRNA5 upregulation is associated with a poor response to radiotherapy in OSCC patients” in accordance with the meaning of our results (Please see Line 218 of the revision).

 Q5: Table 2 also refers to head and neck cancers instead of OSCC. Why do not authors limit its study to oral squamous cell carcinoma (OSCC) as the title proclaims

Re: Thank you for this suggestion. Since the signature of combining high-level CHRNA5 and OSCC type refers to a worse prognosis in Kaplan-Meier analyses against TCGA-HNSC patients as shown in Figure 2, we therefore included HNSC subdivision as a factor in Tables 1 and 2 in order to supporting the findings in Figure 2B.

Q6: There are some typos in the text. For example: line 174 “assoaicates”; Figure 2 legend, lines 195 and 197, head and nech; line 209 “significaly”; line 284, “treatempt”; line 285 “thereapeutic”.

Re: To avoid typos and grammar errors, we have sent our manuscript to American Journal Experts for English editing. The certification is shown as follow.

Reviewer 2 Report

In this manuscript, authors performed an in silico analysis to find the CHRNA5, a gene encoding nicotinic acetylcholine receptor subunit alpha-5 is prominently overexpressed in primary OSCC as compared to normal tissues and serves as an unfavorable risk factor for the survival of OSCC patients. Furthermore, their in vitro data indicated that the endogenous CHRNA5 expression or activation is significantly associated with radioresistance in OSCC cell lines by triggering the E2F activity. They confirmed the prognostic significance of CHRNA5 and E2F gene signature in OSCC receiving radiotherapy. The findings suggest that CGRNA5 plays a significant function in radioresistance of OSCC and might be a pharmaceutical target to enhance the effect of radiotherapy for OSCC. This manuscript is well structured and written. Specific comments to further improve the manuscript are as follows:

The authors should provide the in silico analysis datasets details in the part of Materials and Methods (g., patients cohort and tools for analyses). How to treat the cells with nicotine? Meanwhile, please describe all techniques (e.g., Q-PCR, WB) which have been used in this manuscript. In the survival data (Figure 2), could the authors explain the reduced number of patients as compared to the TCGA-HNSC dataset? Line 169-175, how the authors conclude that CHRNA5 serves as an independent risk factor for the survival rate of HNSCC patients according to table 2? Could the authors explain the reason for the selected 305 patients for Cox univariate and multivariate analyses as well as crosstab analysis? The cut-off value for dividing the low and high expression of CGRNA5? The co-factor Stage should be excluded in multivariate analysis for RFS by Cox proportional hazards model. How is the Kaplan-Meier analysis of CHRNA5 transcript combining E2F gene signature under the condition of overall or disease-specific survival probability? All cell lines used in this manuscript were ever whether confirmed by the Human Cell Line Authentication Test? Line 55-59, there is no literature supporting the statistical data? Line 63, please confirm that Bevacizumab is applied for OSCC in National Comprehensive Cancer Network (NCCN) guidelines. Line79-81, please cite this literature “Indian J Cancer. 2018 Oct-Dec;55(4):399-403. DOI: 10.4103/ijc.IJC_325_18”. Also, discuss the interesting findings in the discussion. Line 110, please note the time point of the cell viability assay after irradiation. In line 182, the authors did not show any data about targeted drug resistance in OSCC? How to explain the disparity of the mRNA levels of CHRNA5 detected by RT-PCR and Q-PCR? Language and style spell check is required (g., Line148, etc).

Author Response

Reviewer 2:

In this manuscript, authors performed an in silico analysis to find the CHRNA5, a gene encoding nicotinic acetylcholine receptor subunit alpha-5 is prominently overexpressed in primary OSCC as compared to normal tissues and serves as an unfavorable risk factor for the survival of OSCC patients. Furthermore, their in vitro data indicated that the endogenous CHRNA5 expression or activation is significantly associated with radioresistance in OSCC cell lines by triggering the E2F activity. They confirmed the prognostic significance of CHRNA5 and E2F gene signature in OSCC receiving radiotherapy. The findings suggest that CGRNA5 plays a significant function in radioresistance of OSCC and might be a pharmaceutical target to enhance the effect of radiotherapy for OSCC. This manuscript is well structured and written. Specific comments to further improve the manuscript are as follows:

The authors should provide the in silico analysis datasets details in the part of Materials and Methods (g., patients cohort and tools for analyses). How to treat the cells with nicotine? Meanwhile, please describe all techniques (e.g., Q-PCR, WB) which have been used in this manuscript. In the survival data (Figure 2), could the authors explain the reduced number of patients as compared to the TCGA-HNSC dataset? Line 169-175, how the authors conclude that CHRNA5 serves as an independent risk factor for the survival rate of HNSCC patients according to table 2? Could the authors explain the reason for the selected 305 patients for Cox univariate and multivariate analyses as well as crosstab analysis? The cut-off value for dividing the low and high expression of CGRNA5? The co-factor Stage should be excluded in multivariate analysis for RFS by Cox proportional hazards model. How is the Kaplan-Meier analysis of CHRNA5 transcript combining E2F gene signature under the condition of overall or disease-specific survival probability? All cell lines used in this manuscript were ever whether confirmed by the Human Cell Line Authentication Test? Line 55-59, there is no literature supporting the statistical data? Line 63, please confirm that Bevacizumab is applied for OSCC in National Comprehensive Cancer Network (NCCN) guidelines. Line79-81, please cite this literature “Indian J Cancer. 2018 Oct-Dec;55(4):399-403. DOI: 10.4103/ijc.IJC_325_18”. Also, discuss the interesting findings in the discussion. Line 110, please note the time point of the cell viability assay after irradiation. In line 182, the authors did not show any data about targeted drug resistance in OSCC? How to explain the disparity of the mRNA levels of CHRNA5 detected by RT-PCR and Q-PCR? Language and style spell check is required (g., Line148, etc). 

Point-by-point responses

Q1: The authors should provide the in silico analysis datasets details in the part of Materials and Methods (g., patients cohort and tools for analyses).

Re: As requested, we have added a paragraph in M&M section and the Supplementary Table 1 of revision to describe the information of TCGA head and neck cancer patients. The content of the paragraph is described as follow.

2.1. TCGA-HNSC patients and data processing

   The clinicopathological data and transcriptional profiling of head and neck cancer patients were obtained from The Cancer Genome Atlas (TCGA) database and downloaded from UCSC Xena website (http://xena.ucsc.edu/welcome-to-ucsc-xena/). The TCGA head and neck cancer patients who have complete clinicopathological information (Tab. S1) were selected in this study. The RNA sequencing results of CHRNA members and E2F geneset deposited in The Molecular Signatures Database (http://software.broadinstitute.org/gsea/msigdb) were downloaded from UCSC Xena website. The E2F signature was generated by using the sum of mRNA levels obtained from the RNA sequencing results of E2F geneset in TCGA head and neck cancer patients. The definition of low and high-levels of CHRNA5 and E2F signature was determined by using Cutoff Finder (http://molpath.charite.de/cutoff/) under the maximal risk condition in Kaplan-Meier analyses.

Q2: How to treat the cells with nicotine? Meanwhile, please describe all techniques (e.g., Q-PCR, WB) which have been used in this manuscript.

Re: Thank you for these comments. We treated the cells with nicotine at the designated concentrations prior to irradiation exposure (Please see Line 245 of the revision). As requested, we have added all techniques, e.g., Q-PCR, WB, utilized in this study in the revised M&M section. The descriptions are as follows.

2.3. Reverse transcription PCR (RT-PCR) and quantitative PCR (Q-PCR)

Total RNA was extracted from cells using a TRIzol extraction kit (Invitrogen). Aliquots (5 μg) of total RNA were treated with M-MLV reverse transcriptase (Invitrogen) and amplified with Taq-polymerase (Protech) using paired primers (for CHRNA5, forward-CGCTCGATTCTATTCGCTACAT and reverse-CAGCACAGTCAAAGGATGAACTTT AC; for GAPDH, forward-AGGTCGGAGTCAACGGATTTG and reverse-GTGATGGCATG GACTGTGGTC) for PCR. Power SYBRTM Green PCR Master Mix (Thermo Fisher) was used for Q-PCR. The mRNA levels were normalized to those of GAPDH. Fold changes were calculated using the 2−ΔΔCt method.

2.4 Western blotting assay

Aliquots (20-100 μg) of total protein and HR Pre-Stained Protein Marker 10-170 kDa (BIOTOOLS, Taiwan) were loaded onto each lane of a SDS gel and transferred to PVDF membranes. The membranes were incubated with blocking buffer (5% nonfat milk in TBS containing 0.1% Tween-20) for 2 hours at room temperature. After that, the samples were incubated with primary antibodies against total/phosphorylated Rb (Cell Signaling) and GAPDH (AbFrontier) overnight at 4 °C. After extensive washing, the membranes were probed with a secondary peroxidase-labeled IgG for 1 hour at room temperature. Immunoreactive bands were visualized by enhanced chemiluminescence (Amersham Bioscience).

2.5 Preparation and infection of lentiviral particle

All derivatives of shRNA vector with puromycin selection marker (obtained from National RNAi Core Facility Platform in Taiwan) were transfected into the packaging cell line 293T (Thermo Fisher) along with the pMD.G and pCMVΔR8.91 plasmids using a calcium phosphate transfection kit (Invitrogen). After 48-hour incubation, the viral supernatants were collected and transferred to the target cells, and the infected cells were cultured in the presence of puromycin (Calbiochem) at 5 - 10 μg/ml in order to selecting the stably transfected cells.

Q3: In the survival data (Figure 2), could the authors explain the reduced number of patients as compared to the TCGA-HNSC dataset?

Re: In Figure 2A, we selected TCGA-HNSC patients who have complete clinicopathological data, overall survival information and RNA sequencing data for CHRNA5. Accordingly, TCGA-HNSC patients who have complete clinicopathological data, recurrence-free survival information and RNA sequencing data for CHRNA5 were selected in Figure 2B. TCGA-HNSC patients who were recorded to receive radiotherapy and have complete clinicopathological data, recurrence-free survival information and RNA sequencing data for CHRNA5 were selected in Figure 2C. Due to these criteria, the number of TCGA-HNSC patients are different in Kaplan-Meier analyses shown in Figure 2.

Q4: Line 169-175, how the authors conclude that CHRNA5 serves as an independent risk factor for the survival rate of HNSCC patients according to table 2? Could the authors explain the reason for the selected 305 patients for Cox univariate and multivariate analyses as well as crosstab analysis? The cut-off value for dividing the low and high expression of CGRNA5? The co-factor Stage should be excluded in multivariate analysis for RFS by Cox proportional hazards model. How is the Kaplan-Meier analysis of CHRNA5 transcript combining E2F gene signature under the condition of overall or disease-specific survival probability?

Re: Thank you for these critical comments. In Table 1, Cox regression test using multivariable analysis reveals that CHRNA5 levels (high v.s. low) compared to other clinical risk factors statistically (p=0.011) predicts an adjusted HR at 1.81 under recurrence-free survival condition. Therefore, we conclude that CHRNA5 serves as an independent prognostic marker for cancer recurrence in HNSCC patients. We have revised the sentences as “Furthermore, regarding recurrence-free survival probability, univariate and multivariate Cox regression analyses revealed that CHRNA5 is an independent prognostic marker with respect to other clinical risk factors that predict cancer recurrence in HNSCC patients (Tab. 1)” to describe the finding in the revised Results section (Please see Lines 205-207 of the revision).

   Because we used the condition of recurrence-free survival probability to perform Cox univariate and multivariate analyses as well as crosstab analysis, as shown in Figure 2B, 305 patients were thus selected in these tests.

   We performed Kaplan-Meier analyses for CHRNA5 under the maximal risk condition in this study. The cut-off value for dividing the low and high expression of CHRNA5 were determined by using Cutoff Finder (http://molpath.charite.de/cutoff/). We have added this information in the 2.1. TCGA-HNSC patients and data processing of revised M&M section (Please see Lines104-106 of the revision)

   As requested, we have removed the co-factor Stage in multivariate analysis for RFS by Cox proportional hazards model in Table 1 (Please see the revised Tab. 1).

   Genes related to the gene set of E2F targets were obtained from The Molecular Signatures Database (http://software.broadinstitute.org/gsea/msigdb). The E2F signature was generated by using the sum of mRNA levels obtained from the RNA sequencing results of E2F gene set in TCGA head and neck cancer patients downloaded from UCSC Xena website and then stratified into high and low-level groups by using Cutoff Finder (http://molpath.charite.de/cutoff/) under the maximal risk condition in Kaplan-Meier analyses. We have described this data processing in the revised M&M section. In Figure 5C, by using this method to define the high and low-level groups for CHRNA5 and E2F gene set, we generated 3 groups regarding low-level CHRNA5 and E2F signature, high-level CHRNA5 and E2F signature and others (low-level CHRNA5/high-level E2F signature and high-level CHRNA5 and low-level E2F signature) in the Kaplan-Meier analysis.

Q5: All cell lines used in this manuscript were ever whether confirmed by the Human Cell Line Authentication Test?

Re: Thank you for this critical comment. These cell lines were purchased from Japanese Collection of Research Bioresources (JCRB) Cell Bank on 2014-11-25 (Please the following document #1) by Dr. Michael Hsiao at Academia Sinica, Taiwan and then stored at liquid nitrogen after 3 passages. All cells were authenticated on the basis of short tandem repeat (STR) analysis, morphologic and growth characteristics and mycoplasma detection by Dr. Michael Hsiao’s Lab as claimed in their previous report (Chang, et. al. Cancer Research 2016). The cell lines used in this study were obtained from their storage at liquid nitrogen. Based on this, we have added sentence “All cells were gift from Dr. Michael Hsiao at Academia Sinica, Taiwan and authenticated on the basis of short tandem repeat (STR) analysis, morphologic and growth characteristics and mycoplasma detection” in the Cell culture paragraph of the revised M&M section. The description is as follow.

2.2. Cell culture

The OSCC cell lines HSC-2, HSC-3, HSC-4 and SAS were cultured in Dulbecco's modified Eagle's medium (DMEM) containing 10% fetal bovine serum (FBS) and 1% nonessential amino acids (NEAA) at 37˚C in a humidified atmosphere containing 7% CO2. All cells were the gift from Dr. Michael Hsiao at Academia Sinica, Taiwan and authenticated on the basis of short tandem repeat (STR) analysis, morphologic and growth characteristics and mycoplasma detection.

Q6: Line 63, please confirm that Bevacizumab is applied for OSCC in National Comprehensive Cancer Network (NCCN) guidelines.

Re: Thank you for this comment. We confirm that Bevacizumab is not applied for OSCC in National Comprehensive Cancer Network (NCCN) guidelines, so that we have removed Bevacizumab in the Introduction section (Please see Line 64 of revision).

Q7: Line79-81, please cite this literature “Indian J Cancer. 2018 Oct-Dec;55(4):399-403. DOI: 10.4103/ijc.IJC_325_18”. Also, discuss the interesting findings in the discussion.

Re: As suggested, we have cited this report as Ref. 13 and also discussed this interesting findings in the Discussion section as a sentence “Previously, CHRNA5 c.1192G>A polymorphism was found to be linked to frequency of tobacco chewing but did not have a significant impact on the risk of OSCC [13]” (Please See Lines 397-399 of the revision).

Q8: Line 110, please note the time point of the cell viability assay after irradiation.

Re: As suggested, we have added the time point “24 hours” of the cell viability assay after irradiation in M&M section (Please see Line 145 of revision).

Q9: In line 182, the authors did not show any data about targeted drug resistance in OSCC?

Re: Thank you for your comment. Indeed, we did not include data about targeted drug resistance in OSCC in this study. So, we removed “targeted drug resistance” and “targeted therapy” in Results section (Please see Lines 218 and 225 of revision).

Q10: How to explain the disparity of the mRNA levels of CHRNA5 detected by RT-PCR and Q-PCR?

Re: Thank you for the critical comment. The disparity of the mRNA levels of CHRNA5 detected by RT-PCR and Q-PCR might be due to the PCR cycle and the resolution of electrophoresis in RT-PCR experiment. Although the trend of relative mRNA levels detected by RT-PCR and Q-PCR are similar, we finally used the results detected by Q-PCR, a modern reliable technique for mRNA quantification, to perform the correlation test in Figure 3C.

Q11: Language and style spell check is required (g., Line148, etc). 

Re: As suggested, we have sent our manuscript to American Journal Experts for English editing. The certification is shown as follow.

Round 2

Reviewer 1 Report

I think that the manuscript may be published in its present form.

Author Response

Reviewer 1

I think that the manuscript may be published in its present form.

Response:

We sincerely thank Reviewer 1 for the great comments and suggestions to this manuscript.

Reviewer 2 Report

1. In Q4: The authors did not response the question.

How is the Kaplan-Meier analysis of CHRNA5 transcript combining E2F gene signature under the condition of overall or disease-specific survival probability?

Otherwise, recurrence-free survival probability data are not sufficient to support the conclusion of prognostic significance of the CHRNA5 transcript level combined with the E2F gene in OSCC.

2. The line numbers between pages are missing in the revised manuscript. The indicated line numbers in Point-by-point letter are not correct.

Author Response

Reviewer 2:

In Q4: The authors did not response the question.

How is the Kaplan-Meier analysis of CHRNA5 transcript combining E2F gene signature under the condition of overall or disease-specific survival probability?

Otherwise, recurrence-free survival probability data are not sufficient to support the conclusion of prognostic significance of the CHRNA5 transcript level combined with the E2F gene in OSCC.

Response:

Thank you for this critical comment. To avoid the misreading in Figs. 5B and 5C, we changed the labels of “E2F signature” to “E2F gene set” in the revised Fig. 5 and revised the figure legend (Please see Lines 314-319 of revision).

In MM section, we revised the sentences “The E2F signature was generated by using the sum of mRNA levels obtained from the RNA sequencing results of E2F geneset in TCGA head and neck cancer patients. The definition of low and high-levels of CHRNA5 and E2F signature was determined by using Cutoff Finder (http://molpath.charite.de/cutoff/) under the maximal risk condition in Kaplan-Meier analyses” to become “The sum of mRNA levels obtained from the RNA sequencing results of E2F gene set in TCGA head and neck cancer patients was used to perform Pearson correlation test and Kaplan-Meier analyses. The stratification of mRNA levels regarding CHRNA5 and E2F gene set into the low and high-level groups was determined by using Cutoff Finder (http://molpath.charite.de/cutoff/) under the maximal risk condition in Kaplan-Meier analyses” in order to more clearly describe the method for performing these analyses (Please see Lines 103-107 of revision).

In Results section, we revised the sentence “In OSCC patients receiving radiotherapy, the gene signature of combined high levels of CHRNA5 and E2F appeared to predict worse prognosis in TCGA OSCC patients receiving radiotherapy (Fig. 5C).” to become “In OSCC patients receiving radiotherapy, the gene signature of combined high levels of CHRNA5 and E2F gene set appeared to predict worse prognosis in TCGA OSCC patients receiving radiotherapy (Fig. 5C).” in order to more clearly describe the data (Please see Line 310 of revision).

The line numbers between pages are missing in the revised manuscript. The indicated line numbers in Point-by-point letter are not correct.

Response:

As suggested, we have carefully checked this issue in the revised manuscript and ensured that the indicated line numbers in Point-by-point letter are correct.
